# Using Fuzzy Neural Networks to Model Landslide Susceptibility at the Shihmen Reservoir Catchment in Taiwan

**Chuen-Ming Huang [1,2,\*], Chyi-Tyi Lee [1], Liu-Xuan Jian [2], Lun-Wei Wei [2], Wei-Chia Chu [3] and Hsi-Hung Lin [3]**

[1] Institute of Applied Geology, National Central University, Taoyuan 320317, Taiwan; ct@gis.geo.ncu.edu.tw
[2] Disaster Prevention Technology Research Center of Sinotech Engineering Consultants, Inc., Taipei 114065, Taiwan; phoenix@sinotech.org.tw (L.-X.J.); lwwei@sinotech.org.tw (L.-W.W.)
[3] Central Geological Survey, Ministry of Economic Affairs, New Taipei 235055, Taiwan; vegachu@moeacgs.gov.tw (W.-C.C.); linch@moeacgs.gov.tw (H.-H.L.)
\* Correspondence: odin@gis.geo.ncu.edu.tw

**Abstract:** Machine learning algorithms are commonly employed in landslide susceptibility assessments. Recently, algorithms that utilize artificial intelligence have come into prominence. This study attempts to adapt the most fundamental framework of deep learning and introduces fuzzy theory concepts to analyze landslide susceptibility while updating the network parameters with trial-and-error methods. The final analysis results will compare with those of logistic regression (LR). In order to assess the ability of the model to identify landslides in a more objective way, two typhoon events were used as a training event and a validation event, respectively. The results of the analysis show that the area under the curve (AUC) of the fuzzy neural network (FNN) for the training event is 0.915, but the AUC for the validation event drops to 0.746. Although the results of the FNN for training events were better than those of LR, they did not differ much from those of LR in predicting future events. The reason for this is that the difference between the landslide distributions of the training and validation events is too large, making the model biased in its identification. Overall, FNN is still a recommended method for analyzing landslide potential and can be used as a reference for LR.

**Keywords:** landslide; landslide susceptibility; artificial neural network; fuzzy; machine learning

## 1. Introduction

Landslide susceptibility analysis has developed over the past few decades. The analysis methods can be broadly classified into two categories, infinite-slope-based and statistically-based methods. Slope stability analysis using infinite slopes [1,2] requires a wide range of material parameters and groundwater level data, which are not easy to collect. As a result, statistical methods are becoming the mainstay of wide-area landslide susceptibility analysis [3].

The term deep learning has emerged rapidly in recent years but its predecessor, machine learning, has been in widespread use for over a decade. In landslide susceptibility analysis the most commonly used machine learning algorithms include logistic regression (LR) [4–6], artificial neural network (ANN) [7–13], decision tree [14–17] and support vector machine [13,18–21]. However, due to a lack of computational power, simple logistic regression is more widely used.

Deep learning is a complex version of artificial neural networks that requires not only a large amount of sample data but also data analysis of the sample data. Therefore, this study uses a simple neural network structure and adjusts its structure and execution parameters by a trial-and-error method. In addition, most of the previous analyses of landslide susceptibility using neural networks set the output layer as a node, output continuous values, and then set threshold values to split into two categories. In order to achieve a more objective classification, this study attempts to combine a fuzzy membership

function [22–26] in the output layer to form a fuzzy neural network (FNN) with a fuzzy function, which has both classification and quantification capabilities.

Over-training happens when using the same landslide inventory and validating model. To avoid this situation, two landslide inventories, generated by typhoon events Aere and Matsa, were used in this study. The landslide inventory of Typhoon Aere is used to train the model and Typhoon Matsa is used to validate the model. In addition, the results of the FNN will be compared with the LR.

## 2. Study Area

We conduct our study in the Shihmen Reservoir catchment, an import water resource in northern Taiwan. The Shihmen Reservoir catchment (757 km$^2$) is steeply dissected, mountainous terrain that includes numerous alluvial terraces (Figure 1). The overall topography is highest in the south and lowest in the north. Elevation ranges from 3500 m at the southernmost edge to about 150 m at the northern end and channels generally flow northwards. In the upland regions of the catchment, high-gradient streams are common and carry enormous amounts of sediment into the Shihmen Reservoir, causing siltation of the reservoir. The average annual rainfall is about 2400 mm. Affected by the plum rain season and typhoons; the main rainy season is from May to October. There are also thunderstorms brought by south-westerly currents and heavy rainfall induced by tropical depressions.

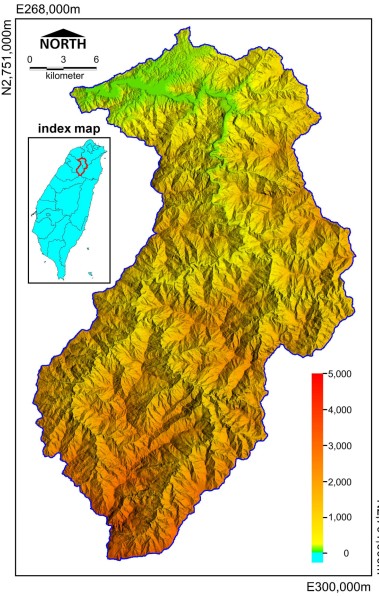

**Figure 1.** Topographic map of the Shihmen Reservoir catchment area in Taiwan.

The geologic setting of the basin is defined by the north-east, south-west trending Chuchih fault, which divides the basin into two different geologic regions. North of Chuchih fault is characterized by sandstone and shale, and south of the fault is dominated by indurate sandstone and shale, argillite, and quartzite. The prevalence of metamorphic rock and the grade of metamorphism gradually increase to the south.

In August 2004, a large number of landslides were triggered in upstream regions of the catchment during Typhoon Aere, which dumped 973 mm of precipitation, with some regions of the basin receiving up to 1600 mm. Massive amounts of soil and mud flushed into the reservoir and increased sediment concentration, disrupting water supply for domestic and industrial use for 17 days. In August 2005, Typhoon Matsa released 819 mm of precipitation in the catchment, which again triggered landslides and disrupted the water supply for 7 days. Though the cumulative rainfall of Typhoon Matsa was less than Typhoon Aere, both storms triggered a similar number of landslides.

## 3. Materials and Methods

### 3.1. Landslide Inventory

Landslide inventories collected after the Typhoon Matsa and Aere events are used to analyze landslide susceptibility in this study. Landslides can be interpreted through pseudo-color images according to the texture, shape and topographical characteristics of the landslide. Prior-event and post-event landslide inventories were created for both storms from the manual interpretation of SPOT-5 satellite images. A sub-sample of the landslide sites was selected randomly to verify locations and boundaries via fieldwork. Through comparison of prior-event and post-event landslide sites (Figure 2), recent landslides and expanded landslides due to the actual event are identified and are named as triggered landslides. Figure 3 shows the triggered landslide inventory of the two events considered in this study.

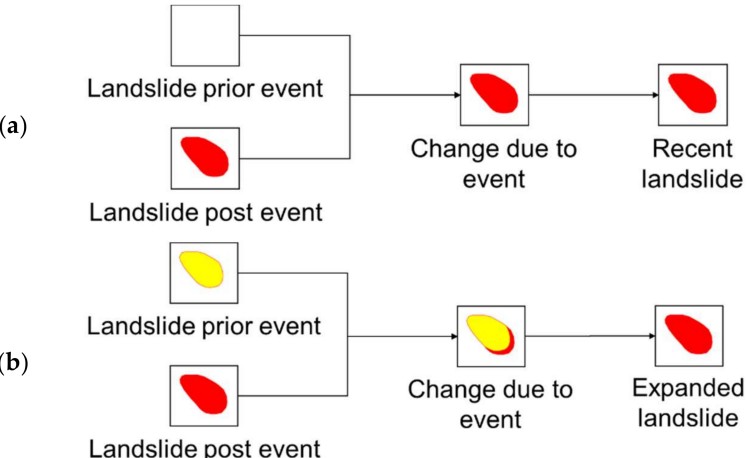

**Figure 2.** The identifying procedure of triggered landslide: (**a**) recent landslide; (**b**) expanded landslide.

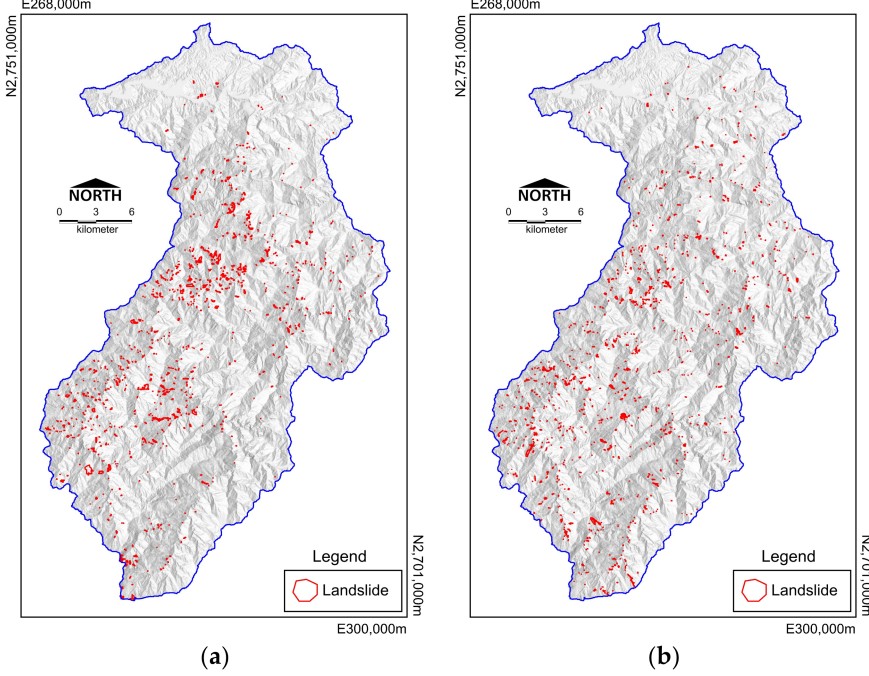

**Figure 3.** The triggered landslide inventory of two events: (**a**) Typhoon Aere; (**b**) Typhoon Matsa.

*3.2. Factor Selection*

There are two types of factors used in this study, the first is a categorical factor, the second one is a continuous factor. Categorical factors are lithology and aspect. These two categorical factors are not screened and are used directly by FNN. The LR is split into 4 different lithology classes and 8 different aspect orientation classes based on 8 directions. Factor selection was carried out for continuous factors. The selection process can be divided into three steps (Figure 4) [4]. First, the available factors in our study area were selected. Second, graphical discrimination was used to identify factors typically needed to correctly interpret a landslide, such as high Area Under Curve (AUC), probability of failure curve matched physical laws, and low overlap between landslide and non-landslide distribution (Figure 5). Finally, high correlated factors were excluded based on correlation analysis.

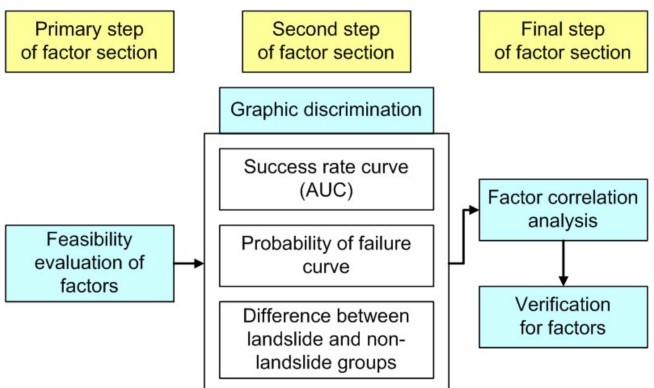

**Figure 4.** The procedure of factor selection.

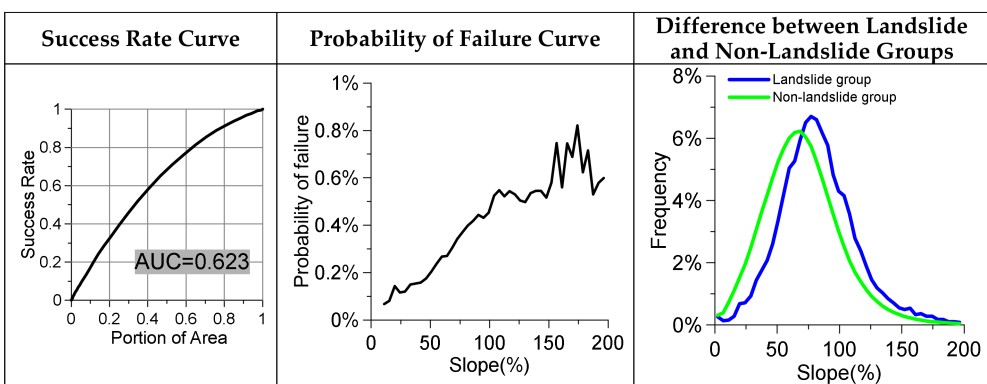

**Figure 5.** The example of graphic discrimination.

This study adopts 12 factors for landslide susceptibility analysis (Table 1). The factors are shown separately in Figure 6. Lithology factors were simplified from a 1:50,000 geology map (Central Geology Survey, Taiwan), to include 4 lithology classes: sandstone and shale, indurated sandstone and shale, argillite, and finally quartzite and argillite. Aspect, slope, slope roughness, tangential curvature and Topographic Wetness Index (TWI) were calculated from a 10 m digital elevation model following Wilson and Gallant [27]. Total slope height and relative slope height were defined as shown in Figure 7 [4]. Distance to a fault was grouped at intervals of 500 m with a maximum of 4000 m in total. The Normalized Difference Vegetation Index (NDVI) was calculated from prior-event SPOT-5 satellite images event using an algorithm based on a near-infrared band and a red band marked as IR and R, separately. Maximum rainfall intensity and total rainfall were interpolated from in situ hourly rainfall maintained by the Central Weather Bureau, Taiwan and Water Resources Agency, Taiwan into grid data. While applying interpolation, the altitude was considered as weight.

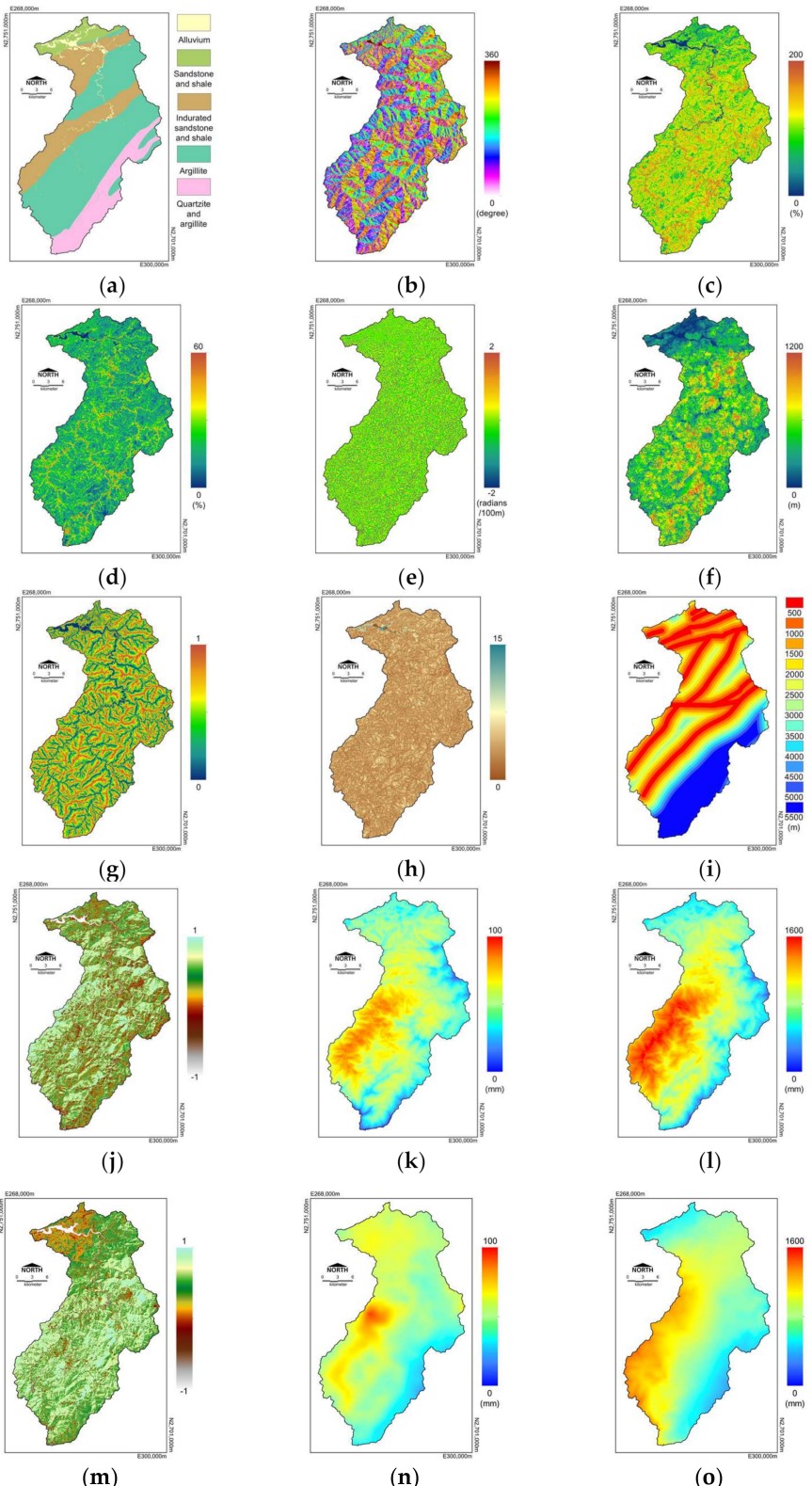

**Figure 6.** The example of selected factors. (**a**) lithology; (**b**) aspect; (**c**) slope; (**d**) slope roughness; (**e**) tangential curvature; (**f**) total slope height; (**g**) relative slope height; (**h**) TWI; (**i**) distance to a fault; (**j**) NDVI of typhoon Aere; (**k**) maximum rainfall intensity of typhoon Aere; (**l**) total rainfall of typhoon Aere; (**m**) NDVI of typhoon Matsa; (**n**) maximum rainfall intensity of typhoon Matsa; (**o**) total rainfall of typhoon Matsa.

**Table 1.** Selected factors for landslide susceptibility.

| Code | Factor Item | Code | Factor Item |
|------|-------------|------|-------------|
| L | lithology | $F_{05}$ | relative slope height |
| A | aspect | $F_{06}$ | TWI |
| $F_{01}$ | slope | $F_{07}$ | distance to a fault |
| $F_{02}$ | slope roughness | $F_{08}$ | NDVI |
| $F_{03}$ | tangential curvature | $F_{09}$ | maximum rainfall intensity |
| $F_{04}$ | total slope height | $F_{10}$ | total rainfall |

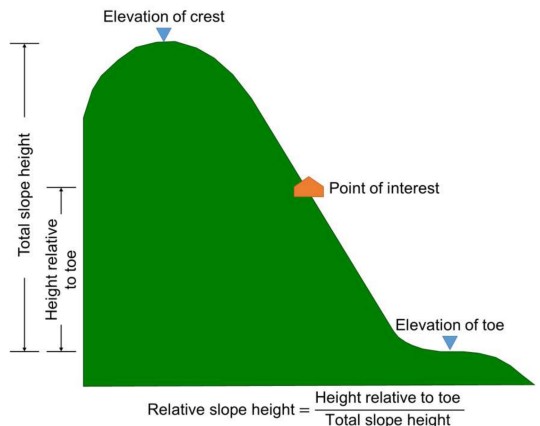

**Figure 7.** The definition of total slope height and relative slope height.

### 3.3. Methodology

3.3.1. Fuzzy Neural Network

Landslide susceptibility analysis is a complex and non-linear problem, and is worth testing by ANN; however, the ANN framework is difficult to quantify and classify at the same time. Typical applications of ANN use an output layer with one node and use a continuous landslide susceptibility value for classification [7,10]. Alternatively, the output layer with two nodes is defined as landslide and non-landslide groups, and the landslide susceptibility is quantified and assigned afterward [12,13]. This study attempted to merge the advantages of fuzzy set theory and the conveniences of ANN to develop a simple neuro-fuzzy model for application to landslide classification and landslide susceptibility analysis [8]. The uniqueness of this model is that two fuzzy membership functions are separately designed for the landslide and non-landslide groups during classification in the output layer (Figure 8). The triangular areas delineated by the fussy membership function have less overlap and the classifying performance is better. The vertical axis of Figure 8 represents the landslide susceptibility, and values 1 to 0 correspond to the possibility of landslide occurrence from high to low.

Hence, the target output of the neural network model is no longer a single node but a series of nodes representing the fuzzy memberships. With this design, the model is capable of classification and quantification. It combines the advantages of ANN and fuzzy set theory while avoiding drawbacks. This model is actually a simple FNN and can be carried out in Matlab.

In this study, trial and error was used to decide the number of hidden layers and the number of nodes in the hidden layer. The number of nodes in the input layers coincides with the number of landslide causative factors and a triggering factor. The number of nodes in the output layers is the same as the designed nodes of the fuzzy memberships for landslides and non-landslides. In this way, the framework of FNN was established. This framework is similar to a common ANN, which is capable of lessening the error between the predicted output values and the calculated output values.

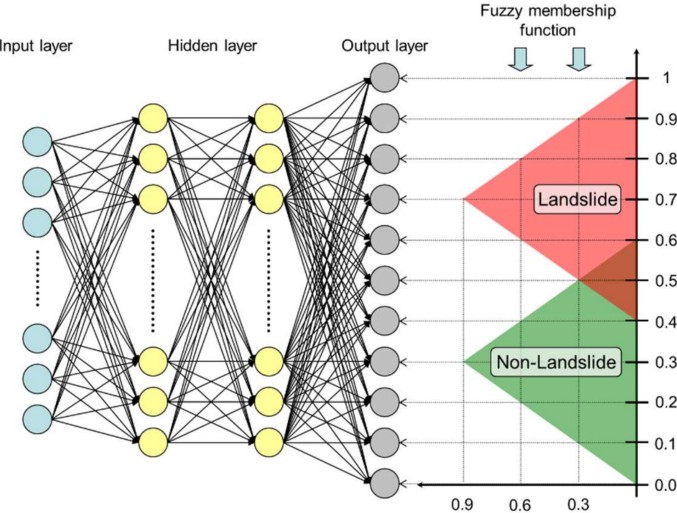

**Figure 8.** The framework of FNN.

Ideally, the model should be trained with a large number of data, so that the connection weights between the nodes are stabilized and the error is minimized. The output is a series of values representing a fuzzy membership. The degree of fuzzy neartude (fuzzy degree) to a landslide group or a non-landslide group may be measured by comparing the output fuzzy set with a membership function. This comparison may be carried out using the "clustering" technique outlined below [28]:

$$S_i = \sum_{i=1}^{n} \mu_i * \alpha_i \tag{1}$$

$$W_i = \frac{S_i}{\sum\limits_{i=1}^{m} S_i} \tag{2}$$

where $S$ is the relevance of the origin series belonging to the target series, $\mu$ is the target series, $\alpha$ is the origin series, and $W$ is the relevant coefficient origin series belonging to the target series. Finally, max ($W$) is used to determine the classification.

Defuzzyfication is the step that transforms output linguistic variables into crisp values. Crisp values are Landslide Susceptibility Index (LSI) in this study. The most common defuzzyfication method is the Centre of Gravity method given by:

$$y = \frac{\int \mu_b(x) x dx}{\int \mu_b(x) dx} \tag{3}$$

where $y$ is the output fuzzy set and $\mu_b$ is a membership function.

### 3.3.2. Logistic Regression

LR is a common method used in traditional statistical analysis. It is often considered the entry point to machine learning and is a common algorithm for classification problems. Compared to linear analysis, the linear relation between dependent variables and independent variables is not necessary, and the format of independent variables is not limited. Dependent variables in binary format are suitable for LR analysis which is usually used to analyze landslide susceptibility [4–6] to overcome the various and complex formats of geological data. The equation is as follows:

$$\ln\left(\frac{P}{1-P}\right) = \sum_{i=1}^{m} w_i F_i + C \tag{4}$$

where $P$ is the 'dependent variable', in this case, the LSI, calculated as the relative incidence given the independent variable $F$ (i.e., each factor), $w$ is the coefficient of the respective independent variable, and $C$ is a constant term. Regression was performed on the training data to obtain the respective independent variable coefficient $w$ and constants $C$. When regressed, the landslide group $P$ is 1 and the non-landslide group $P$ is 0. The LSI for each cell is calculated after bringing all parameters into Equation (4).

*3.4. Model Calibration*

All analyzed data was converted to a 10 m grid. The study area was then divided into stable areas and unstable areas that included landslide and non-landslide according to the topographic character. The stable areas are flat, or the slope gradient percentage is under 10% and the area is over 1 ha. Stable areas typically included river channels and/or flat regions, where landslide initialization is not likely to happen and were thus not used in the analysis. The number of non-landslide samples was greater than the number of landslide samples. The same amount of non-landslide samples was randomly selected for training in order to reduce the deviation. If the non-landslide sample was around a landslide object or was once a landslide, it was not selected.

The coefficient of Equation (4) will be obtained through these training samples. The analyzed results of the logistic regression are as follows:

$$
\begin{aligned}
\ln\left(\frac{P}{1-P}\right) = 1.541 \quad & L_1 + 2.797L_2 + 2.777L_3 + 1.362L_4 - 0.221A_1 + 0.323A_2 \\
& + 1.377A_3 + 1.472A_4 + 1.595A_5 + 1.179A_6 + 0.341A_7 \\
& + 0.001A_8 + 0.377F_{01} + 0.035F_{02} + 0.307F_{03} + 0.180F_{04} \\
& - 0.670F_{05} + 0.040F_{06} - 0.024F_{07} - 0.596F_{08} + 0.612F_{09} \\
& + 0.678F_{10} - 4.588
\end{aligned}
\tag{5}
$$

The dependent variable $p$ is taken as LSI in this study and its range is from 0 to 1. $L_1$ represents sandstone and shale units, $L_2$ represents indurated sandstone and shale units, $L_3$ represents argillite units, and $L_4$ represents quartzite and argillite units. $A_1$ is the aspect direction within the range 337.5 to 22.5, $A_2$ is the aspect direction within the range 22.5 to 67.5, $A_3$ is the aspect direction within the range 67.5 to 112.5, $A_4$ is the aspect direction within the range 112.5 to 157.5, $A_5$ is the aspect direction within the range 157.5 to 202.5, $A_6$ is the aspect direction within the range 202.5 to 247.5, $A_7$ is the aspect direction within the range 247.5 to 292.5, and $A_8$ is the aspect direction within the range 292.5 to 337.5. Table 1 shows the names of factors $F_{01}$ to $F_{10}$.

Same training samples will be used to calibrate FNN model. In this study, the number of hidden layers and nodes in a hidden layer were determined by trial and error. The learning rate and momentum term were set to be 0.01 and 1, respectively, whereas the initial weights were randomly selected. The number of epochs was set to 500 and the root mean square error (RMSE) goal for stopping was set to 0.0001. Most of the iterations stopped within the 500 epochs. However, if the goal met the 0.0001 RMSE goal, the iteration was stopped.

During model calibration, the best framework ($12 \times 30 \times 30 \times 11$) was obtained. The framework is shown in Figure 8, the input layer has 12 nodes, the hidden layer has 2 layers and each layer has 30 nodes, and the output layer has 11 nodes. It produced a fuzzy membership function for each grid. Using the defuzzy method, the fuzzy membership function could be transferred to a continuous value known as the LSI. This study defined the value LSI as between 0 and 1. Through the grey clustering technique [28], landslide groups and non-landslide groups could be classified at the same time.

## 4. Results

During model calibration, the optimum FNN and LR model was obtained, and it produced a fuzzy membership function for each grid. Through the grey clustering technique [28], the fuzzy membership function was divided into a landslide group and a non-landslide group.

Classified accuracy is a common method that identifies the quality of a model. Landslide accuracy, non-landslide accuracy, and overall accuracy is the measure of performance of FNN in this study. The classified result of the training event was displayed through a confusion matrix (Table 2) and formulas of accuracy. In the training phase (Table 3), all accuracies are over 80% and landslide accuracy achieves 90%. All validated accuracies are over 70%, which shows that the model can predict landslide occurrence well.

**Table 2.** Confusion matrix.

|  |  | Observed | |
| --- | --- | --- | --- |
|  |  | **Landslide** | **Non-Landslide** |
| Predicted | Landslide | a | b |
|  | Non-Landslide | c | d |

Landslide accuracy: a/(a + c). Non-Landslide accuracy: d/(b + d). Overall accuracy: (a + d)/(a + b + c + d).

**Table 3.** Classified result of training event and validated event.

|  | Aere Event (Training) | | Matsa Event (Validation) | |
| --- | --- | --- | --- | --- |
|  | **FNN** | **LR** | **FNN** | **LR** |
| Landslide Accuracy | 90.1% | 82.1% | 75.6% | 72.7% |
| Non-Landslide Accuracy | 80.2% | 78.4% | 71.3% | 74.0% |
| Overall Accuracy | 84.7% | 80.8% | 72.9% | 73.6% |

The success rate curve (SRC) and prediction rate curve (PRC) can be plotted by actual landslide and LSI [29]. The AUC is used as a measurement of the FNN model reliability. The baseline of the AUC value is 0.5 and if the AUC value is close to 1, the capability of the FNN model to interpret a landslide is considered good. Figure 9 shows that the FNN model performs well. Although the validation result was inferior to the training result, it is still close to 75%. AUC and the accuracy of the validated event show stable results. Therefore, FNN is shown to be useful in landslide susceptibility analysis. Thus, the landslide susceptibility analysis results were reasonable and acceptable.

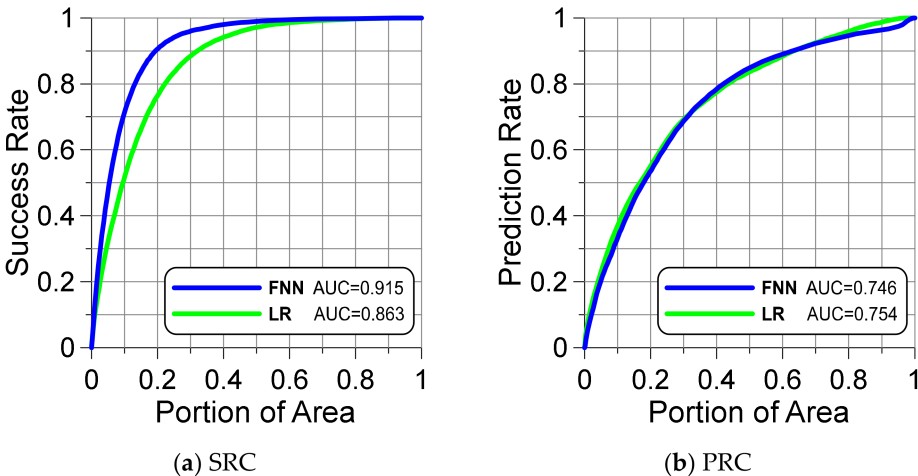

(**a**) SRC　　　　　　　　　　(**b**) PRC

**Figure 9.** The performance of model: (**a**) training event; (**b**) validated event.

The LSI domain was divided into intervals. For each interval, we counted the landslide pixel amount and total pixel amount. The probability of failure was computed by dividing the calculated landslide pixel amount by the total pixel amount. The probability of failure and LSI was used to plot the curve (Figure 10) and fit. A trend was obtained from the probability of failure curve. High LSI indicates a high probability of failure, and the LSI translates to a probability of failure that indicates the spatial probability of landslide occurrence when plotted over the catchment area (Figure 11).

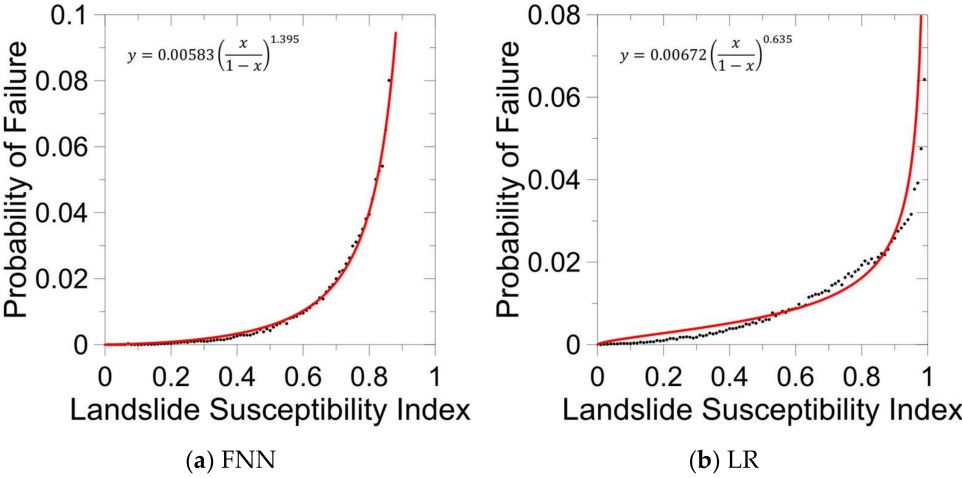

(**a**) FNN           (**b**) LR

**Figure 10.** The result of curve fitting: (**a**) FNN's probability of failure curve; (**b**) LR's probability of failure curve.

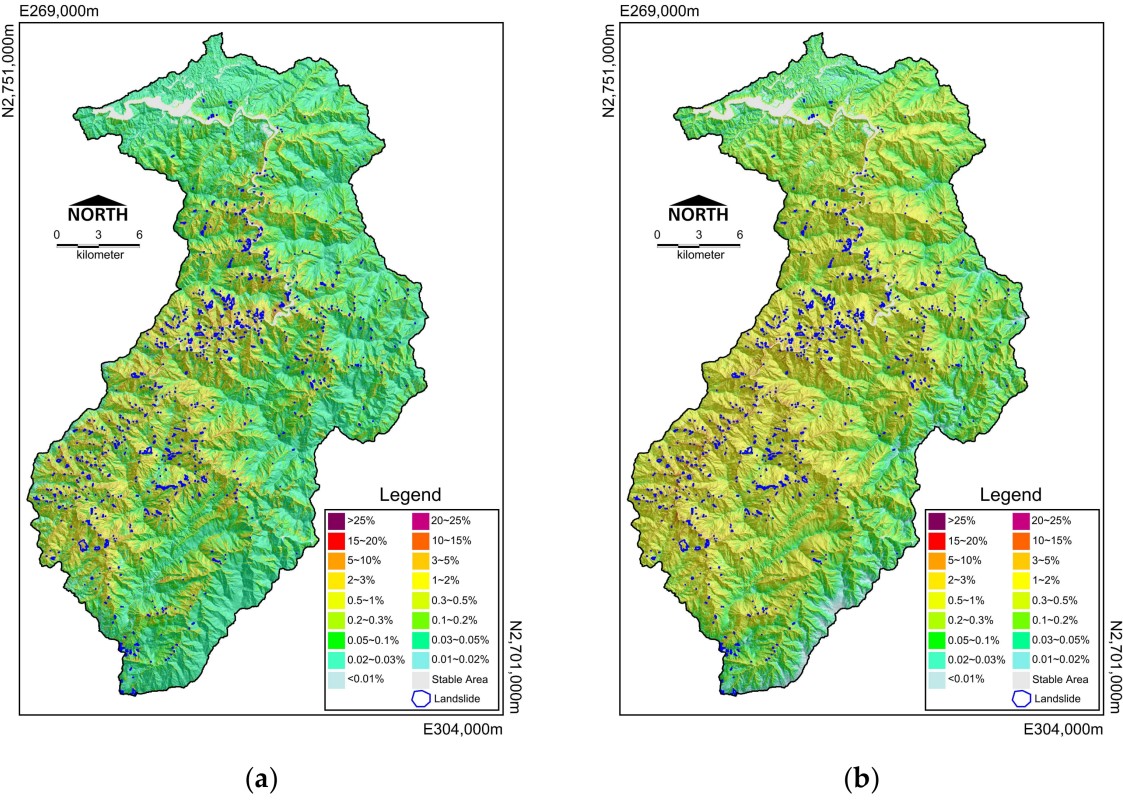

(**a**)                         (**b**)

**Figure 11.** *Cont*.

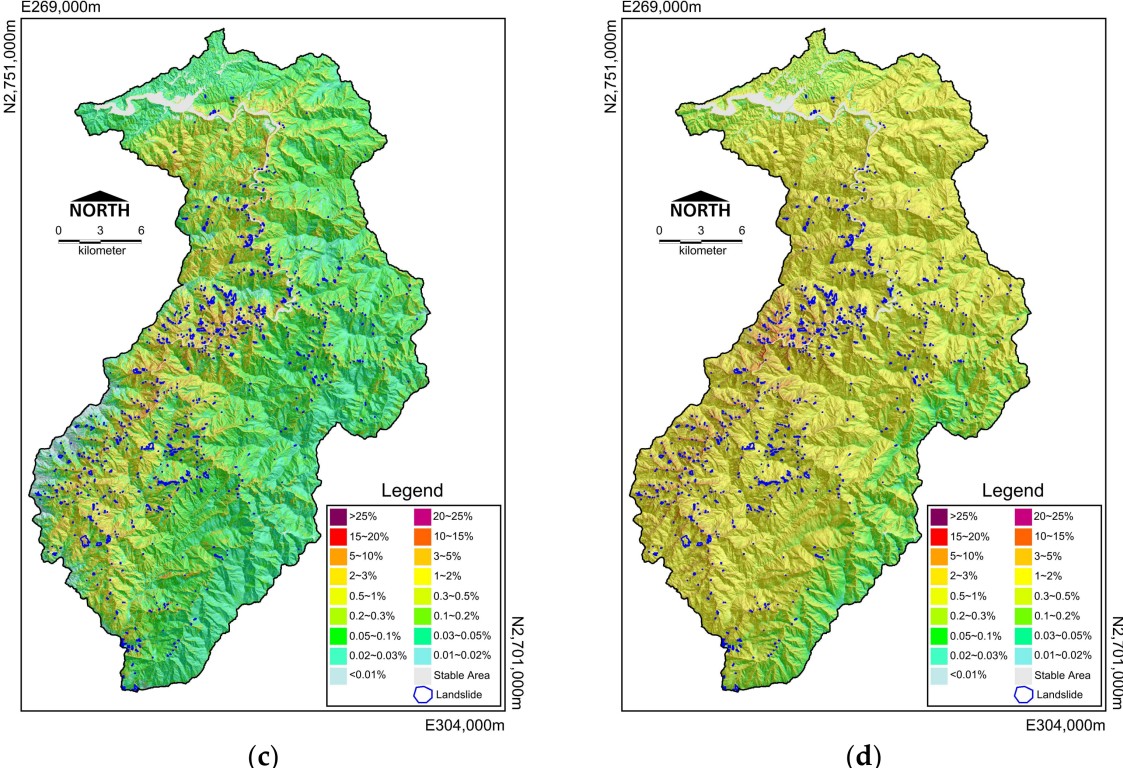

(**c**)                                                    (**d**)

**Figure 11.** Landslide probability map. (**a**) training result of FNN; (**b**) training result of LR; (**c**) validation result of FNN; (**d**) validation result of LR.

## 5. Discussion

The framework of ANN and training parameters such as the number of hidden layers, number of hidden nodes, learning rates, momentum terms, and number of training cycles are acquired via an empirical process of performing sensitivity analyses. Generally, trial and error is the most familiar method employed. Although the gradient descent algorithm can increase analysis efficiency, it still requires a large amount of training time and often takes more time before ANN can obtain optimal parameters.

ANN is also a statistical approach that requires model calibration. The difference is that ANN can characterize nonlinear, nonparametric, or hierarchical mapping functions and does not require information that connects input and output variables if landslide sampling is adequate. During the process of model calibration, underfitting and overfitting are familiar problems. Due to the principle of ANN, if the training cycle and factors are adequate, underfitting is much less common, however, overfitting easily occurs compared with the statistical approach, and in particular, the amount of training sampling is less.

The statistical approach often uses cross-validation, early stopping, or model comparison to avoid unvalidated data. Some researchers used the same samples to train and validate a landslide susceptibility model, but the model's reaction for future events may not be adequate [30]. Chung and Fabbri (2008) [31] proposed dividing the study area into sections such as left and right: one for training and the other for validation. However, they found model performance was best if there were independent training and validation data. Additionally, the model may lose the training pattern if a particular geological region of the model does not match the training pattern.

This study used separate landslide-triggering typhoon events for training and validating so as to avoid overfitting. However, the validation result of FNN looks like another kind of overfitting (see Table 3 and Figure 9). The training and validating results of FNN were similar (Figure 11a,c). Since parts of landslides used for validation were distributed in the area that received very little precipitation, this led to poor validation performance. Incomplete data dominated the validation performance rather than overfitting in this case.

The more accurate the prediction of the future events; the more landslide inventories are needed.

This study attempted to compare the LR of landslide susceptibility and proved that FNN is suitable. However, landslide susceptibility actually gives a relative probability for a region. Landslide susceptibility from different models shows that landslide occurrence possibility is not equal. Therefore, the result of LR and FNN are only able to compare with each other when they are in the same base. To achieve the goal, landslide susceptibility is transferred to landslide probability in this study. This study attempted to fit a function between landslide susceptibility and probability of failure (see Figure 10). Through the function, landslide susceptibility is transferred to a landslide probability that is a kind of spatial probability. The meaning is the region of particular susceptibility had the same occurrence possibility and we do not know where it is.

After converting the landslide susceptibility into spatial probabilities, the landslide probability of FNN and LR have the same basis of comparison. The fact that the LR probability map had more areas of high interstitial probability in training and validation results (Figure 11b,d) made the training accuracy lower. Due to this reason, the impact of incomplete data is obscure compared to FNN. However, there are also more areas to focus on, increasing the cost of disaster prevention.

Regardless of training and validation, FNN and LR have good classification accuracy. All predicted results were inferior to training results when the trained model was verified. FNN performed better in landslide accuracy and overall accuracy but it had the opposite outcome in non-landslide accuracy. A comparison of the SRC and PRC shows that their difference was small. LR is a recognized and stable statistical approach.

Compared with LR, the analysis capacity of FNN is close to LR and FNN has the additional benefits of coping with the data in vast numbers and with no causal relationship. Therefore, if an analysis has the option of using LR or FNN for landslide susceptibility analysis, we suggest using FNN.

## 6. Conclusions

One of the greatest limitations of ANNs has been that the analysis process is too time-consuming, and that the architecture and computational parameters must be determined by trial and error. Additionally, overfitting also often happens in ANN. Although statistical analysis costs considerable time in optimizing analysis samples, it is much faster than ANN. However, with advances in technology and increased computing power, analyzing small data is no longer a problem.

This study uses FNN to evaluate landslide susceptibility. The results indicate that the FNN produced satisfactory results in terms of the overall accuracy and the PRC. Although we found that the predictive capability of FNN was not much better than LR, FNN provided an alternative method for landslide classification. Overfitting can occur in both the LR and FNN methods. The only way to solve this problem is by collecting more event-triggered landslide inventories and developing a complete training dataset.

When the differences in sample data are small, (i.e., the sample data have different value domains, vary in shape (type), and have no causal relationship) an ANN may be attempted to determine landslide susceptibility. Alternatively, it could be used as an ancillary reference to the statistical approach. ANN may not be useful for large landslide datasets due to the computational overhead of optimizing the training dataset.

**Author Contributions:** Conceptualization, C.-T.L.; Data curation, C.-M.H.; Formal analysis, C.-M.H.; Funding acquisition, W.-C.C. and H.-H.L.; Methodology, C.-M.H.; Project administration, C.-T.L.; Validation, C.-T.L. and L.-W.W.; Writing–original draft, C.-M.H.; Writing–review & editing, L.-X.J. and L.-W.W. All authors have read and agreed to the published version of the manuscript.

**Funding:** This research did not require external funding.

**Institutional Review Board Statement:** Not applicable.

**Informed Consent Statement:** Not applicable.

**Data Availability Statement:** All data can be found in the text.

**Conflicts of Interest:** The authors declare no conflict of interest.

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
