# Peer review of "Using Fuzzy Neural Networks to Model Landslide Susceptibility at the Shihmen Reservoir Catchment in Taiwan"

_water, doi:10.3390/w14081196_

Round 1

Reviewer 1 Report

I read the manuscript. The subject is interesting and the idea seems sound. However, the presentation of the manuscript should be improved considering the following comments:
1) The language of the manuscript should be polished.
2) The location names are not given precisely. For ex. Figure 1 tells nothing. Is this Taiwan?
3) The authors mention two typhoons. However, their location and dates are not given. Also, their consequences must be explained.
4) I could not detect information about landslides. Some characteristics features of the landslides must be given.
5) Geology of the study area must be given.
6) How the authors decide the output membership functions and the degree of overlap of two membership functions?
7) Conclusion part is a little bit weak and can be enhanced.

Reviewer 2 Report

The paper appears interesting and generally well structured. The following aspects should be improved:

1) Use of English (minor spell checking is required; see attached file);

2) Some sentences are not clear at all (see details in the attached file);

3) Some quantities need to be defined;

4) The process of data acquisition and, above all, of use of original data in both LR and FNN methods are not satisfactorily explained;

5) Layout of Figure 6 must be improved.

Further specific comments are reported in the attached annotated copy.

Author Response

This manuscript is a resubmission of an earlier submission. The following is a list of the peer review reports and author responses from that submission.

Round 1

Reviewer 1 Report

Thank you for the interested work

Reviewer 2 Report

Please see the marked copy of the manuscript and revise accordingly

Reviewer 3 Report

Dear Authors,

Although the topic is interesting and falls within the scope of the journal, I see several issues with the manuscript:

  1. The presentation is weak. The literature and the methods are not sufficiently explained. The training data are not clearly defined. The results are not properly presented and discussed.
  2. The recent landslide susceptibility studies with other methods, such as decision trees, SVM, etc., provide higher accuracy values. The advantages of the proposed methodology need to be discussed considering the lower prediction accuracy.
  3. The legend of Figure 7 is not quite clear. The areas with higher landslide probability are questionable especially for the LR method.